# Fractional Vegetation Cover Derived from UAV and Sentinel-2 Imagery as a Proxy for In Situ FAPAR in a Dense Mixed-Coniferous Forest?

**Birgitta Putzenlechner** [1,2,*] , **Philip Marzahn** [2,3] , **Philipp Koal** [4] and **Arturo Sánchez-Azofeifa** [5]

1    Institute of Geography, Georg-August-University, Goldschmidtstr. 5, 37077 Göttingen, Germany
2    Department of Geography, Ludwig-Maximilians-University, Luisenstr. 37, 80333 Munich, Germany; philip.marzahn@uni-rostock.de
3    Geodesy and Geoinformatics, University of Rostock, Justus-von-Liebig-Weg 6, 18054 Rostock, Germany
4    Forestry Research and Competence Center ThüringenForst, Jägerstr. 1, 99867 Gotha, Germany; philipp.koal@forst.thueringen.de
5    Earth and Atmospheric Sciences Department, University of Alberta, 1-26 Earth Sciences Building, Edmonton, AB T6G2E3, Canada; arturo.sanchez@ualberta.ca
*    Correspondence: birgitta.putzenlechner@uni-goettingen.de

**Abstract:** The fraction of absorbed photosynthetic active radiation (FAPAR) is an essential climate variable for assessing the productivity of ecosystems. Satellite remote sensing provides spatially distributed FAPAR products, but their accurate and efficient validation is challenging in forest environments. As the FAPAR is linked to the canopy structure, it may be approximated by the fractional vegetation cover (FCOVER) under the assumption that incoming radiation is either absorbed or passed through gaps in the canopy. With FCOVER being easier to retrieve, FAPAR validation activities could benefit from a priori information on FCOVER. Spatially distributed FCOVER is available from satellite remote sensing or can be retrieved from imagery of Unmanned Aerial Vehicles (UAVs) at a centimetric resolution. We investigated remote sensing-derived FCOVER as a proxy for in situ FAPAR in a dense mixed-coniferous forest, considering both absolute values and spatiotemporal variability. Therefore, direct FAPAR measurements, acquired with a Wireless Sensor Network, were related to FCOVER derived from UAV and Sentinel-2 (S2) imagery at different seasons. The results indicated that spatially aggregated UAV-derived FCOVER was close (RMSE = 0.02) to in situ FAPAR during the peak vegetation period when the canopy was almost closed. The S2 FCOVER product underestimated both the in situ FAPAR and UAV-derived FCOVER (RMSE > 0.3), which we attributed to the generic nature of the retrieval algorithm and the coarser resolution of the product. We concluded that UAV-derived FCOVER may be used as a proxy for direct FAPAR measurements in dense canopies. As another key finding, the spatial variability of the FCOVER consistently surpassed that of the in situ FAPAR, which was also well-reflected in the S2 FAPAR and FCOVER products. We recommend integrating this experimental finding as consistency criteria in the context of ECV quality assessments. To facilitate the FAPAR sampling activities, we further suggest assessing the spatial variability of UAV-derived FCOVER to benchmark sampling sizes for in situ FAPAR measurements. Finally, our study contributes to refining the FAPAR sampling protocols needed for the validation and improvement of FAPAR estimates in forest environments.

**Keywords:** essential climate variable; ECV; FAPAR; FCOVER; forest; Sentinel-2; UAV

## 1. Introduction

The fraction of absorbed photosynthetic active radiation (FAPAR) links PAR to the absorption of plants and characterizes the vegetation structure, as well as the energy and carbon cycle [1,2]. FAPAR is a crucial input for climate and ecological models to monitor the net productivity of ecosystems and assess global carbon balances [3,4]. For its key role in

ecosystem processes and productivity, it is listed as one of the terrestrial Essential Climate Variables (ECVs) by the GCOS [5,6]. Satellite remote sensing provides spatially distributed information on vegetation spectral properties, and numerous retrieval algorithms have been developed to retrieve FAPAR from space [7–12]. Thus, there is an increasing availability of FAPAR datasets from hectometric (Terra & Aqua/MODIS [13], SPOT/VEGETATION [14], PROBA-V [15] and Sentinel-3/OLCI [16]) to decametric (Sentinel-2/MSI [17]) spatial resolutions. Most of these products are available at relatively coarse spatial resolutions, thereby suitable only for regional-to-global FAPAR studies. For most agricultural and forestry applications, such as vegetation health, productivity and drought monitoring, higher spatial resolution is required. Efforts have been made to downscale products by using auxiliary optical remote sensing observations [18,19]. Furthermore, with the availability of a harmonized Sentinel-2 and Landsat-8 surface reflectance product [20], new approaches for determining vegetation parameters are expected [21].

Accurate estimates of FAPAR are important to ensure the reliable assessment of the carbon cycle. According to GCOS [5,6], an accuracy of the maximum of 10% and 0.05 is considered acceptable for FAPAR products. The current satellite-derived FAPAR products are close to fulfilling the accuracy requirements, but further improvements are still needed [22,23]. In this regard, several studies continue to report lower product quality in forest ecosystems than in other biomes [22,24–28]. Further, it is emphasized that there is a need for high-resolution satellite data to monitor forest conditions and productivity [29,30], so further efforts should be made for product development, validation and improvement at forested study sites. According to Bayat et al. [23], FAPAR satellite and ground products have not yet reached product maturity, such as, e.g., LAI products. Product maturity can be characterized by the availability of long-term global data records from both satellite and in situ observations, as well as community-agreed sampling and validation protocols, typically developed by the Committee on Earth Observation Satellites (CEOS) within the subgroup of Land Product Validation (LPV) [23]. In contrast to LAI, CEOS LPV has not published a good practice document on FAPAR yet. This situation may be attributed to the fact that FAPAR ground measurements may incorporate considerable uncertainties, so commonly-agreed sampling protocols for FAPAR are still under development [23,31]. Nevertheless, FAPAR ground data is urgently needed, which is typically scarce in forest ecosystems [31].

The methods for monitoring FAPAR comprise direct and indirect methods. FAPAR measurements can be carried out with direct PAR measurements, for example, arranged in Wireless Sensor Networks (WSNs). Two-flux measurements based on incoming and transmitted PAR have been found to provide efficient and accurate ground truth data under typical summer conditions [32]. So far, only a few sites are equipped to provide FAPAR estimates in forest ecosystems, and often, validation studies have to rely on few radiation measurements [22]. A representative sample size is a crucial issue in forests, as the variability of FAPAR varies considerably with the illumination conditions across different ecosystems and within single forest stands [33–35], so that multiple, synchronized sampling is required [36]. In sum, direct FAPAR measurements require considerable time and effort, so that FAPAR is often determined indirectly based on structural information of vegetation [37].

Numerous studies with the aim of validating satellite products at a larger scale have used indirect retrieval methods such as measurements of gap fractions retrieved from digital hemispherical photography (DHP) [38–40], LAI [25,41] or estimates of fractional vegetation cover (FCOVER) [42,43]. The link between the FAPAR and structural canopy parameters is based on the fact that the total area of foliage mainly determines how much incoming solar radiation is intercepted by the canopy so that the FAPAR generally increases with the leaf cover [34]. A closely related quantity to FAPAR is fractional vegetation cover (FCOVER), which is defined as the proportion of horizontal vegetated area occupied by the vertical projection of canopy elements [44]. FAPAR is approximated with FCOVER under the assumption that a little scattering of PAR takes place in green, healthy vegetation,

meaning that incoming PAR is either absorbed or passes through gaps in the canopy [43]. To be more precise, this approximation refers to the sometimes used term "FIPAR", which relates to the fraction of incident radiation reaching the background level based on directly transmitted light only (i.e., without any scattering events inside the canopy) [45].

Information on FCOVER is required for research many land surface processes, climate change and numerical weather predictions, and it is an important variable for many applications in agriculture, forestry, resource and environmental management, land use, hydrology, disaster risk management and drought monitoring [46]. FCOVER is retrieved from optical satellite remote sensing imagery using mainly empirical models (e.g., based on NDVI) or machine learning approaches [46]. A large number of satellite remote sensing products are available, such as FCOVER retrieved from SPOT/VEGETATION [15], Terra & Aqua MODIS [47] (i.e., percent tree cover) and LANDSAT/TM&ETM+ [48], as well as the S2 FCOVER product at 10-m spatial resolution [17]. In field conditions, FCOVER can be retrieved easier than FAPAR with digital (hemispherical) photography. Recently, Unmanned Aerial Vehicles (UAVs) have been used to derive vegetation coverage and gap patterns from multispectral reflectance data and 3D point clouds [49,50]. FCOVER retrieval from UAV imagery has mostly been achieved by means of empirical relationships of multispectral or RGB reflectance and vegetation indices (VIs) with field samples [51,52], RGB or multispectral image classification [53,54] and color unmixing techniques [55]. In forests, FCOVER retrieval based on classification of the spectral information is complicated by the occurrence of dark, shadowed areas between tree crowns [53]. Thus, other methods have been applied, such as the manual digitalization of canopy gaps [56] or the application of ground classification algorithms based on the elevation information of point clouds obtained from structure-from-motion photogrammetry (SfM) [57].

Evaluating the availability of the vegetation parameters FAPAR and FCOVER from remote sensing, it can be noted that FCOVER products are better available. In addition, the validation of FAPAR products often relies on indirect FAPAR measurement methods based on structural vegetation parameters [43,58]. We therefore see a need to better understand the link between FCOVER and FAPAR in field conditions, for example, by exploring the potential of UAV-derived FCOVER to approximate the FAPAR ground measurements. FA-PAR is a quantity that is rather difficult to acquire in a representative way due to the high variability of forests' light environment [33,36,59], so UAV-derived FCOVER could be useful for planning direct and indirect (e.g., DHP) FAPAR sampling campaigns. In addition, we are currently not aware of an evaluation of both ground and remote sensing FAPAR and FCOVER products in forests. Despite the increasing availability of satellite remote sensing products, the current products lack consistent investigations and validation against ground data, especially in forests. Studies on UAV-derived FCOVER mostly focus on crops [55,60], tree plantations [52,54,57] or low-growing vegetation [51], so evaluations in forests are currently underrepresented [53]. Further, there are few studies on the evaluation of UAV-derived FCOVER with decametric satellite remote sensing products like the S2 FCOVER product, which again focuses on non-forested areas [51,54]. As for the S2 FCOVER product, it has been validated in crops [39] or several biomes, including deciduous forests [61], but to our knowledge, it has not been evaluated with UAV-derived FCOVER data at a forest site yet.

In this study we investigated the hypothesis that FCOVER serves as a proxy for in situ FAPAR. Thus, the overall aim is to explore the consistency of remotely sensed FCOVER and in situ FAPAR, referring to both absolute values and spatial variability. Our specific objectives were thus to

- retrieve FCOVER from UAV flights for different phenological periods,
- evaluate the consistency between the S2 FAPAR and FCOVER products based on high-resolution ground observations and discuss the potential of remotely sensed FCOVER to approximate in situ FAPAR and its spatial variability.

We also discuss the potential of UAVs for efficient FCOVER retrieval by considering uncertainties due to data quality and ground mask retrieval. The assessment contributes to

the further understanding of the relationship between FAPAR and FCOVER in relatively closed forest stands, which can be used to develop sampling protocols for FAPAR.

## 2. Materials and Methods

### 2.1. Study Site

The experiment was carried out at the subalpine site "Graswang" (47.5708°N, 11.0326°E; 864 m altitude) in Southern Germany, which is part of the pre-Alpine TERENO research project for long-term environmental research [62]. The climate is warm temperate and fully humid, with the vegetation period spanning from late April to the end of September. The site is in a valley bottom and comprises a mid-aged spruce-dominated mixed-coniferous forest. According to a forest inventory from 2015, the forest stand comprises 82% (of the basal area) Norway spruce (*Picea abies*), which is a typical species in South German commercial timberlands. The deciduous species are European beech (*Fagus sylvatica*, 14%) and Sycamore maple (*Acer pseudoplatanus*, 4%). The forest floor is composed of a low-growing understory (mainly *Oxalis acetosella* and *Mercurialis perennis*) that appears green throughout the vegetation period. The trees have an average height of 15 m, the mean DBH is 0.13 m and the stem density and basal area account to 231 and 4.8 m$^2$ per ha, respectively. The stand is relatively closed, with only a few larger gaps between tree crowns (Figure 1a). Table 1 shows the overall basal area and species composition in the surrounding sensor locations for PAR measurements in the forest.

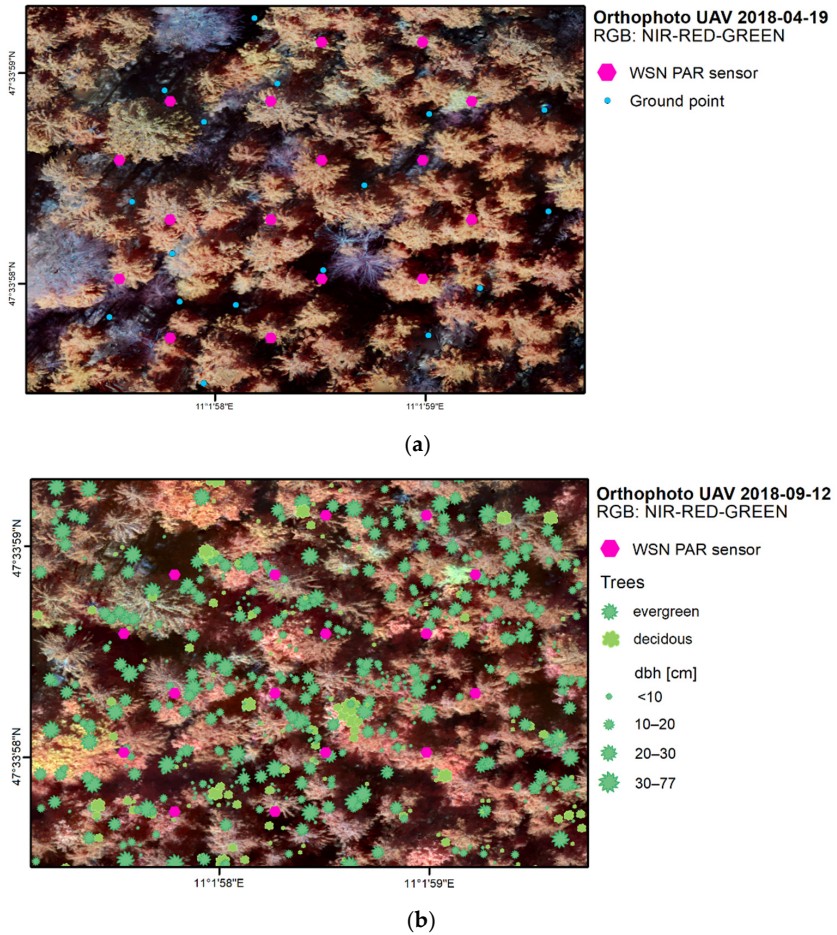

**Figure 1.** Overview of the area covered by the Wireless Sensor Network of PAR sensors (pink hexagons) for FAPAR measurements. UAV orthophotos are shown in a false color infrared band combination: (**a**) image taken during defoliated deciduous canopy in the spring; the blue point symbols indicate seed points needed for the ground classification routine; (**b**) image taken in the summer with foliated deciduous canopy; green symbols represent the DBH and species composition.

**Table 1.** Species composition in the 10-m surroundings of the 16 sensor locations (for measurements of transmitted PAR). The small letters in the column "location ID" indicate spruce-dominated (coniferous, "c") or mixed (coniferous–deciduous, "d") species compositions.

| Location ID | Basal Area (m$^2$ ha$^{-1}$) | Basal Area of Conifers (%) | Species Composition |
|:---:|:---:|:---:|:---:|
| 2c | 0.7 | 94.3 | conifer-dominated |
| 3c | 0.5 | 96.7 | conifer-dominated |
| 4c | 0.4 | 95.5 | conifer-dominated |
| 5d | 0.5 | 79.2 | mixed |
| 6d | 0.4 | 65.9 | mixed |
| 7d | 0.5 | 82.1 | mixed |
| 8c | 0.4 | 93.1 | conifer-dominated |
| 9c | 0.5 | 91.9 | conifer-dominated |
| 10d | 0.6 | 82.0 | mixed |
| 11d | 0.7 | 81.4 | mixed |
| 12d | 0.4 | 77.7 | mixed |
| 13c | 0.4 | 95.1 | conifer-dominated |
| 14c | 0.5 | 92.2 | conifer-dominated |
| 15d | 0.5 | 79.4 | mixed |
| 16d | 0.5 | 81.7 | mixed |

*2.2. FAPAR Measurements*

2.2.1. Experimental Design and Instrumentation

The experimental set-up for the FAPAR measurements consisted of a WSN of self-powered nodes (ENV-Link-Mini-LXRS, LORD MicroStrain, Cary, NC, USA) deployed in a hexagonal topology (Figure 1). The sampling scheme resembles a grid-based sampling approach with the advantage of maximizing the sensing area while ensuring signal connectivity [63,64]. The nodes were deployed with a spacing of 10 m to account for the average footprint of the PAR measurement, which is dependent on tree height (see Reference [28] for further details). The PAR measurements were carried out by commercially available quantum PAR sensors (SQ-110, Apogee, Logan, UT, USA; field of view 180°) mounted 1.3 m high on wooden poles and directed upward. Inside the forest, 16 sensors measured transmitted PAR ($PAR_{trans}$), covering an area of approx. 0.2 ha. A reference sensor outside the forest on open grassland measured the incoming PAR ($PAR_{in}$). The nodes were configured to perform measurements every 10 min synchronously. Preprocessing of the PAR data included a restriction to daylight timesteps, a terrain shadow exclusion routine to account for the influence of the nearby slopes on the measurements and an exclusion of data acquired during mixed illumination conditions based on measurements of the TERENO meteorological station (see Reference [35] for further details).

2.2.2. Processing of Two-Flux FAPAR Estimates

Based on $PAR_{in}$ and $PAR_{trans}$, a two-flux *FAPAR* estimate was calculated. Thus, no additional sensors and towers were required. This estimate ignored the influences of horizontal *PAR* fluxes, as well as top-of-canopy and *PAR* albedo of the ground surface. Despite the underlying assumptions, the two-flux estimate has been shown to deliver accurate estimates of *FAPAR* (bias < 0.05) under typical summer conditions or around solar noon when the effect of the solar zenith angle is lowest [32,36].

*FAPAR* was calculated at each sensor location in the forest ("individual *FAPAR*") following Equation (1) and as the spatial average thereof ("domain *FAPAR*") following Equation (2):

$$FAPAR_{i,t} = 1 - \frac{PAR_{trans_{i,t}}}{PAR_{in_t}} \tag{1}$$

$$FAPAR_{n,t} = \frac{1}{n} \sum_{i}^{n} FAPAR_{i,t} \tag{2}$$

where $FAPAR_{i,t}$ is the individual *FAPAR* at time step $t$ at an individual sensor location $i$, and $FAPAR_{n,t}$ is the domain mean *FAPAR* at time step $t$ based on all $n$ sensor locations.

The resulting time series was filtered for time steps $\pm 1$ h around solar noon to both limit the influence of the solar zenith angle on the accuracy of *FAPAR* and match the acquisition times of the S2 and UAV images to ensure similar solar constellations. As satellite imagery is only meaningful on cloudless days, we filtered the data for clear sky conditions by using the ratio of diffuse-to-total incident radiation ($d/Q < 0.2$) of a sunshine pyranometer of a meteorological station (see Reference [35] for further details). To account for the slightly different acquisition dates of the remote sensing imagery, domain and individual *FAPAR* were analyzed, including $\pm 10$ days around the UAV and S2 acquisition dates. It should be noted that our *FAPAR* estimate refers to the concept of "black sky *FAPAR*", which is also widely used for the retrieval of *FAPAR* remote sensing products [6,15].

### 2.3. UAV Imagery

### 2.3.1. UAV and Camera System

The UAV platform consisted of a Rotorkonzept-RKM 8X octocopter (RotorKonzept GmbH, Abtsteinach, Germany), which is a lightweight UAV for payloads up to 1.1 kg. Depending on the wind conditions and payload, the flight duration was around 12 min. Flights were controlled by a Pixhawk flight controller to fly with a nominal speed of $5 \text{ ms}^{-1}$ at 100 m above the canopy with a forward overlap of 80% and 65% side overlap between single images. The UAV was equipped with a multispectral camera (RedEdge, MicaSense, Seattle, WA, USA) to acquire images with 12-bit depth, stored as five 16-bit TIFF files per channel. A calibration procedure to convert from radiance to reflectance was carried out according to the recommendations of the manufacturer. Before the flights, a calibration target with known spectral albedo was imaged, with the camera assuming constant conditions of the atmosphere during the flight. A gimbal with brushless engines was used to correct for yaw and roll effects to ensure nadir-looking imaging geometry.

### 2.3.2. Image Acquisition and Postprocessing

Flights were carried out close to solar noon (i.e., 12:30 p.m.) in the early spring (19 April 2019) and late summer (12 September 2018) to acquire images when deciduous trees were leafless and with full foliage with similar sun positions. The solar zenith and azimuth angles at 12:30 p.m. were 37° and 33°, as well as 186° and 184° for the two dates, respectively. The UAV flights covered an area of approx. 2.5 ha, including the area of the WSN (Figure 2). To enhance the geometric accuracy of the images, seven ground control points (GCPs) were placed as markers for postprocessing throughout the acquired area and measured using a differential GPS (model Trimble R2, Trimble Inc., Sunnyvale, CA, USA) with an average horizontal error of 0.43 m and an average vertical error of 1.15 m. The markers were left in the field throughout the year to ensure the same geometric quality for both acquisition dates.

The images were postprocessed with Agisoft Photoscan software (v. 1.3.2, Agisoft LLC, St. Petersburg, Russia) to derive dense point clouds with SfM techniques. Orthomosaics and digital surface models (DSM) were generated and exported as TIFF files only for visualization purposes (Figures 1 and 2). As the herbal understory is green in the summer, ground classification methods based on the threshold, as often applied for crops [60], were not applicable. To avoid this problem, we followed the recommendation of Yao et al. [50] to exploit the geometric information of UAV data by using the elevation information of the 3D point cloud for the retrieval of ground masks.

Therefore, a ground classification with previous noise removal was performed using the operator "Classify Ground" within the graphical modeling tool "Spatial Modeler" within Erdas Imagine software (v. 16.6.0.2100, Erdas Inc./Hexagon, Atlanta, GA, USA). The implemented algorithm uses an iterative process to classify the input point cloud into ground and non-ground points based on a ground classification approach known as adaptive TIN (triangulated irregular network) [65]. In the parameterization of the

algorithm, we noticed that the use of ground control points was needed to generate the provisional TIN, which was then iteratively densified by computing the distance of each unclassified point from the developing ground surface. Points with a vertical distance from the surface with less than or equal to 0.25 m ("offset value") were classified as ground points and iteratively added to the ground surface. For a robust ground classification result, the configuration included an input of 137 ground control points as seed points for the initial TIN, which were verified in the field and by visual inspection of the orthomosaics. After the ground classification, the thematic point cloud was converted to a raster of 0.05-m spatial resolution. Focal majority filtering (5 × 5 matrix) was applied to fill the small data gaps. However, some data gaps remained in dark, shadowed areas. Considering different possible class affiliations of these data gaps ("NA"), we retrieved and analyzed two ground masks by assuming either NA as ground or NA as vegetation. Thus, two FCOVER products were obtained (FCOVER-UAV$_{NA=ground}$ and FCOVER-UAV$_{NA=veg}$) from the ratio of ground to overall pixels within the footprints of the PAR measurements (see Section 2.5).

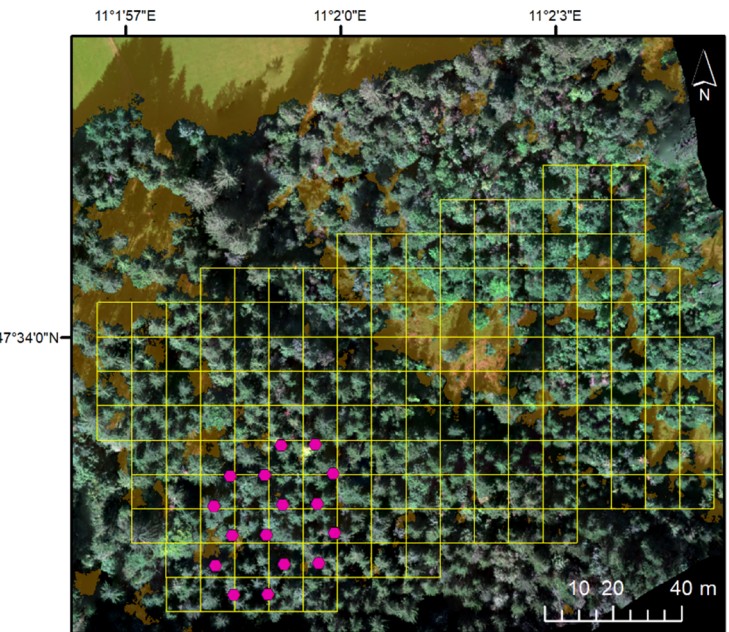

**Figure 2.** Spatial extents of remote sensing and in situ products: UAV orthophoto from 12 September 2018 with WSN (pink hexagons), footprints of S2 pixels (yellow outlines) and UAV-derived ground mask (assumption NA = vegetation, shown in transparent brown) for FCOVER retrieval.

### 2.4. Sentinel-2 FAPAR and FCOVER

The Sentinel-2 mission by ESA with a constellation of two identical satellites provides multispectral data at a decametric resolution with 2 to 3 days revisit time at mid-latitudes [66]. Cloud-free Level 2A data was downloaded from the Copernicus Open Access Hub for two acquisition dates: 19 April 2018 and 16 September 2018 to represent the early vegetation period (deciduous trees leafless) and full foliated canopy, respectively. The Level 2A data represented top-of-canopy reflectance and were been corrected for atmospheric and topographic effects using the "Sen2Cor" algorithm [67]. The Level 2B biophysical products of FAPAR and FCOVER were retrieved using the "Biophysical Processor" within the Sentinel-2 toolbox of the ESA software SNAP (https://step.esa.int/main/download/snap-download/, accessed on 11 January 2022). We used the newly implemented version 2.0 (available since SNAP 8.0), which provides a refined version of the algorithm for biophysical parameter retrievals (i.e., LAI, FAPAR and FCOVER) based on the 10-m bands [17]. The improved spatial resolution of this implementation in comparison to a previous version (20 m) supports a better match to high-resolution UAV images. The results were checked on inconspicuous quality indicators.

The generic algorithm implemented in the "Biophysical Processor" is based on a neuronal network trained over the PROSPECT + SAIL radiative transfer model on top-of-canopy reflectance data without prior information on the land cover [17]. The product definition of FAPAR corresponds to daily integrated "black sky" FAPAR, which correspond roughly to measurements under clear sky conditions around the local solar noon. Further, the algorithm refers to "green FAPAR" and "green fractional vegetation cover", which means that non-green vegetation elements are not considered.

### 2.5. Spatial Linking of Products

The coordinate reference system of UAV and Sentinel-2 imagery was WGS 84, UTM 32N (EPSG: 32632). A linking methodology was required to compare products of different spatial resolutions and extents (Figure 2). The products were evaluated twofold: (a) at single sensor locations of the WSN, referring to the footprints of the PAR sensors (Section 2.5.1) and, (b) to a larger extent, of the whole area flown based on S2 pixels (only applicable for S2 and UAV products) (Section 2.5.2).

### 2.5.1. Linking of Remote Sensing Products to In-Situ FAPAR

Remote sensing products were linked to in situ FAPAR measurements based on the footprints of the PAR measurements. Therefore, standard GIS routines (Erdas Imagine) were used to calculate buffer areas with a radius of 10 m around all $i$ sensor locations for $PAR_{trans}$. UAV-derived FCOVER was calculated from the ratio of ground-to-non-ground pixels within the respective buffer area following Equation (3) and averaged for the whole area following Equation (4), which could then be compared to individual and domain FAPAR, respectively.

$$FCOVER - UAV_i = 1 - \frac{\text{number of ground pixels}}{\text{total number of pixels}} \tag{3}$$

$$FCOVER - UAV_n = \frac{1}{n} \sum_{i}^{n} \cdot FCOVER - UAV_n \tag{4}$$

Due to the coarser spatial resolution of S2, the method needed further adaption for *FAPAR-S*2 and *FCOVER-S*2. Therefore, all $m_i$ intersecting S2 pixels $j_i$ for the footprint area of a sensor location $i$ were identified (and then also for all $n$ sensor locations of the FAPAR WSN). Based on the proportion share $w_{j_i}$, the S2 *FCOVER* (and FAPAR) values were calculated as the weighted average at single sensor locations following Equation (5), thus corresponding to individual FAPAR (Equation (1)). *FCOVER-S*2 (and *FAPAR-S*2) values corresponding to the FAPAR domain (Equation (2)) were calculated following Equation (6) (for a schematic illustration of the linking method, see Reference [28]).

$$FCOVER - S2_i = \sum_{j_i}^{m_i} w_{j_i} \cdot FCOVER - S2_{j_i} \tag{5}$$

$$FCOVER - S2_n = \frac{1}{n} \sum_{i}^{n} \cdot FCOVER - S2_i \tag{6}$$

### 2.5.2. Linking Remote Sensing Products at Larger Extent

Since the area covered by the WSN was considerably smaller than the whole area flown, we additionally evaluated the S2 products (i.e., *FCOVER-S*2 with *FCOVER-UAV*) at the largest possible spatial extent. Therefore, the remote sensing products were compared based on the S2 pixels over an area of approx. 1.6 ha (Figure 2). To compare *FCOVER-UAV* to *FAPAR-S*2 and *FCOVER-S*2, *FCOVER-UAV* was calculated based on the ground masks following Equations (3) and (4) (in this case, referring to individual pixels $i$ of all pixels $n$).

*2.6. Statistical Analysis*

Different UAV ground masks were compared visually regarding their total number and size of gaps. We calculated the ratios of "small gaps" (<5 m$^2$) and "very small gaps" (<1 m$^2$) thereof based on the classification scheme used by Reference [56].

The comparison between FAPAR ground measurements and remote sensing products was carried out at individual sensor locations. Therefore, a two-sample Kolmogorov–Smirnov (KS) test was applied to test on the null hypothesis of similar value distributions between individual FAPAR-WSN, FAPAR-S2, FCOVER-S2 and FCOVER-UAV on the 95% significance level. Combinations for which the null hypothesis could not be rejected ($p > 0.05$) are indicated in the figures with small letters. To evaluate the product agreement on seasonal dynamics, the differences between the spring and summer acquisition dates were calculated. The linear relationship and total agreement of the products was assessed with the following metrics: the coefficient of determination ($R^2$), root mean square error (RMSE) and percent bias (%bias, as the average amount that FAPAR-WSN is greater than FCOVER-UAV as a percentage of the absolute value of FAPAR-WSN). We differentiated for the sensor locations where the ratio of the basal area of conifers was below 85% (Table 1).

To compare the spatial variability of all products, we calculated the maximum coefficient of variation in percent as a function of the number of samples taken at the 16 sensor locations. Therefore, all possible combinations were considered with the binomial coefficient (as $n = 16$, choose k combinations). We depict the calculated coefficient of variation as a function of the number of samples taken and evaluate it with a 5% uncertainty threshold that can be seen as the most conservative product uncertainty requirement when referring to the GCOS accuracy recommendation for FAPAR products (i.e., the maximum of 10% and 0.05). The resulting dataset was checked for linear relationships between different remote sensing products and in situ FAPAR measurements.

## 3. Results

*3.1. Remote Sensing Products*

3.1.1. FCOVER-UAV

The UAV-derived FCOVER (FCOVER-UAV) was calculated based on a ground classification of the point cloud elevation information. Figure 3 shows the resulting ground masks for both acquisition dates. It is evident that generally more gaps are delineated at the spring, compared to the summer, acquisition date. Thus, despite the dominance of conifers at this site, the phenology of the deciduous trees had a major effect on the number and size of the gaps, irrespective of the assumptions on data gaps. In the summer, the canopy is almost closed, and only few larger gaps can be seen. However, the ground masks showed considerable differences, depending on the underlying assumptions on the data gaps. Ground masks only coincided with larger gaps where the canopy was less dense (Figure 3).

Table 2 summarizes the influence of the ground mask on the number and size of the gaps, as well as derived FCOVER. Assuming NA as vegetation reduced the number of delineated gaps drastically, and assuming NA as ground resulted in a larger number of gaps. Differences in the number of gaps depending on the ground mask applied were particularly high at the summer (Figure 3b), and less pronounced at the spring, acquisition date (Figure 3a). In fact, assuming NA as vegetation in the summer resulted in 93% less gaps than assuming NA as ground; in the spring, assuming NA as vegetation resulted in 75% less gaps (Table 2). It is further striking that the delineated gaps are smaller on average for the assumption of NA as ground. This is also seen in the higher occurrence of "very small gaps" (<1 m$^2$): assuming NA as ground resulted in up to 92% of "small gaps" (<5 m$^2$) in the spring compared to 68% in the summer when assuming NA as vegetation (Table 2).

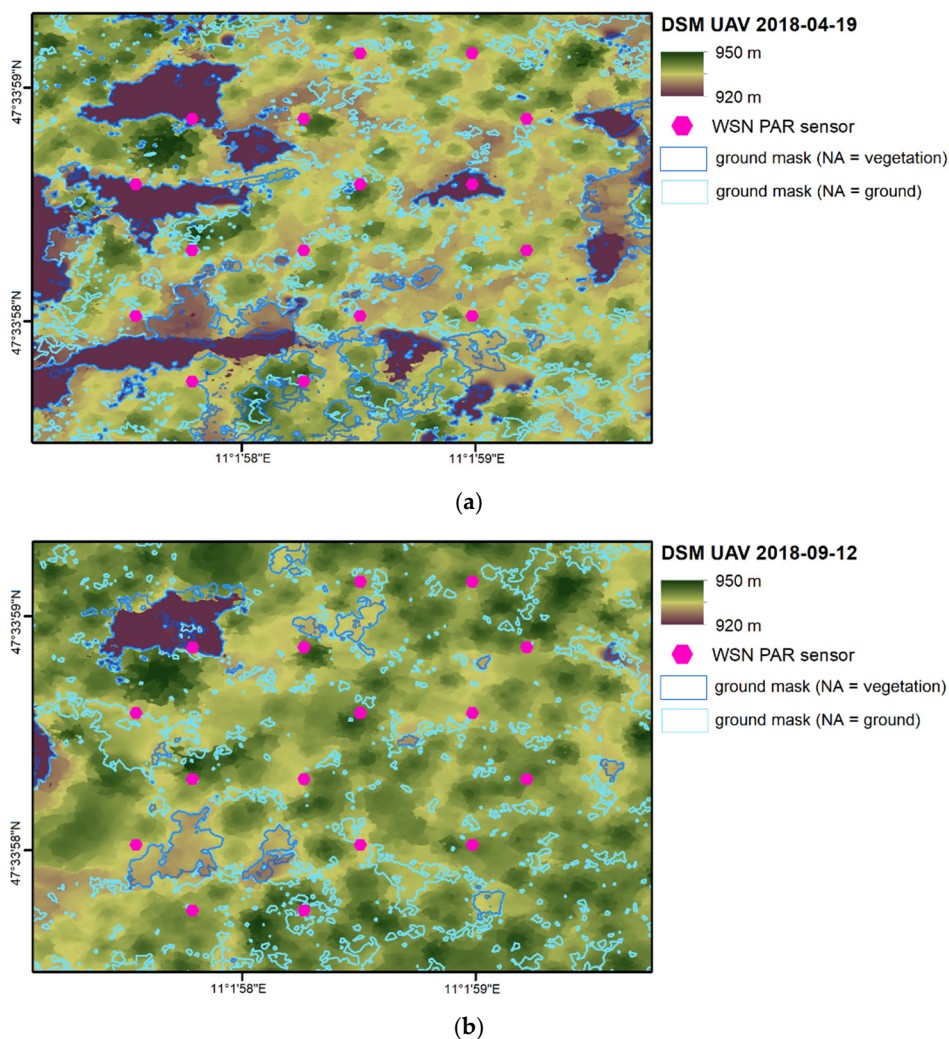

**Figure 3.** Results of ground classifications at different acquisition dates compared to DSMs. Ground masks derived from point clouds are shown in light blue and dark blue outlines for the (**a**) spring and (**b**) summer acquisition dates.

**Table 2.** Gap numbers, sizes and derived FCOVER-UAV depending on the assumptions in ground classification.

|  | Defoliated Canopy | | Foliated Canopy | |
| --- | --- | --- | --- | --- |
|  | NA = Ground | NA = Vegetation | NA = Ground | NA = Vegetation |
| FCOVER-UAV | 0.20 | 0.64 | 0.63 | 0.94 |
| number of gaps | 515 | 129 | 414 | 27 |
| Gap size: mean $\pm$ sd (m$^2$) | $4.3 \pm 68.4$ | $10.5 \pm 85.3$ | $2.5 \pm 17.9$ | $6.5 \pm 16.9$ |
| Ratio of small gaps (<5 m$^2$) (%) | 97 | 91 | 95 | 81 |
| of which very small gaps (<1 m$^2$) (%) | 92 | 93 | 90 | 68 |

Differences of the ground masks were also reflected in the FCOVER-UAV. For the spatial extent of the WSN, the seasonal differences ranged between 0.30 (FCOVER-UAV$_{NA=ground}$) and 0.43 (FCOVER-UAV$_{NA=veg}$), corresponding to seasonal differences of 33% and 67% (Table 2). Figure 4 shows that this relative difference is also observed over a larger spatial extent (Figure 2). With generally less area delineated as ground in the spring, the value distributions differed considerably: in the spring, FCOVER-UAV$_{NA=veg}$ was distributed

almost evenly over the entire range of values (Figure 4a), while, in the summer, the majority of the values fell between 0.8 and 1 (Figure 4b).

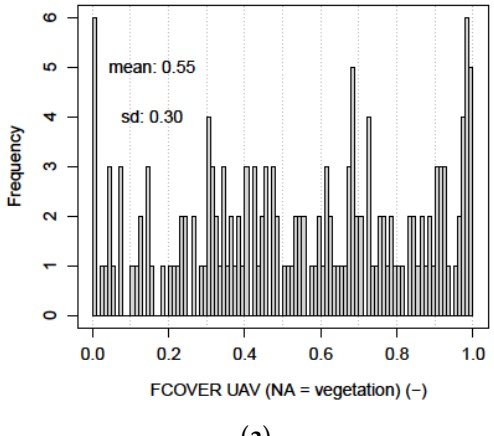
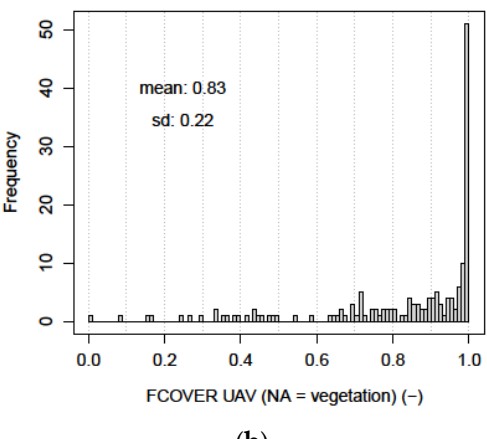

(**a**)                         (**b**)

**Figure 4.** Value distributions of FCOVER-UAV$_{NA=veg}$ across the study site for (**a**) the spring and (**b**) summer acquisition dates. FCOVER was derived on the basis of S2 pixels.

### 3.1.2. Sentinel-2 Products

The FAPAR (FAPAR-S2) and FCOVER (FCOVER-S2) were retrieved on 19 April and 16 September 2018. On average, FCOVER-S2 across the site accounted for 0.4 in the spring and 0.6 in the summer, corresponding to a seasonal difference of 30% (Figure 5b,d). In comparison, the values of FAPAR-S2 were around 0.2 significantly higher ($p \leq 0.05$), ranging from 0.6 in the spring and 0.8 in the summer (Figure 5a,c). Thus, the seasonal differences of FAPAR-S2 were slightly below (25%) FCOVER-S2. The value distributions of FAPAR-S2 and FCOVER-S2 were significantly different ($p \leq 0.05$). The most obvious differences in the summer were the rather narrow-shaped value distributions of FAPAR-S2 and a higher standard deviation of FCOVER-S2 (0.14 vs. 0.09). Overall, Figure 6 shows that the products are closely related ($R^2 > 0.96$), especially in the higher value range of the two quantities.

### 3.1.3. Evaluation of FCOVER-S2 with FCOVER-UAV

To quantify the level of agreement between FCOVER-S2 and FCOVER-UAV, Figure 7 shows scatterplots from the FCOVER calculations from the UAV ground mask based on S2 pixels. While FCOVER-UAV spans over the whole range of values in the spring, FCOVER-S2 ranges from 0.2 to 0.6. In sum, FCOVER-S2 is only around half as FCOVER-UAV, with a mean percent bias ranging from 40% to 50% for the spring and summer acquisition dates, respectively. Already, the visual interpretation of the orthophoto suggests that FCOVER-S2 underestimates FCOVER at this site. Although the canopy is particularly dense in the south and southwestern parts of the site and the area of the WSN (Figure 1), the pixels of FCOVER-S2 show values up to 0.6 (Figure 8), which can be seen in the comparisons between Figures 1b and 8b.

Figure 7 also shows that, while a weak correlation is obtained at the spring acquisition date ($R^2 = 0.2$), the observations are uncorrelated in the summer and lost under saturation. A visualization of FCOVER-S2 and the ground masks underlying FCOVER-UAV reveals further details of this seasonally different agreement. In this regard, Figure 8 shows UAV-derived ground masks and FCOVER-S2 for the area of the WSN where the forest is relatively dense and FCOVER-S2 should reach saturation in the summer. However, the delineation of gaps in the UAV-derived ground mask is not reflected in FCOVER-S2, showing values around 0.6 at locations that are outlined as gaps. Contrarily, in the spring, higher values of FCOVER-S2 occur in the central and southeastern parts of the area, where a high coverage has been identified by the UAV-derived ground mask. Thus, the visual product comparison confirms that the agreement between the FCOVER-S2 and UAV-derived ground mask

(particularly assuming NA as vegetation) is higher for the spring (Figure 8a) than the summer acquisition date (Figure 8b).

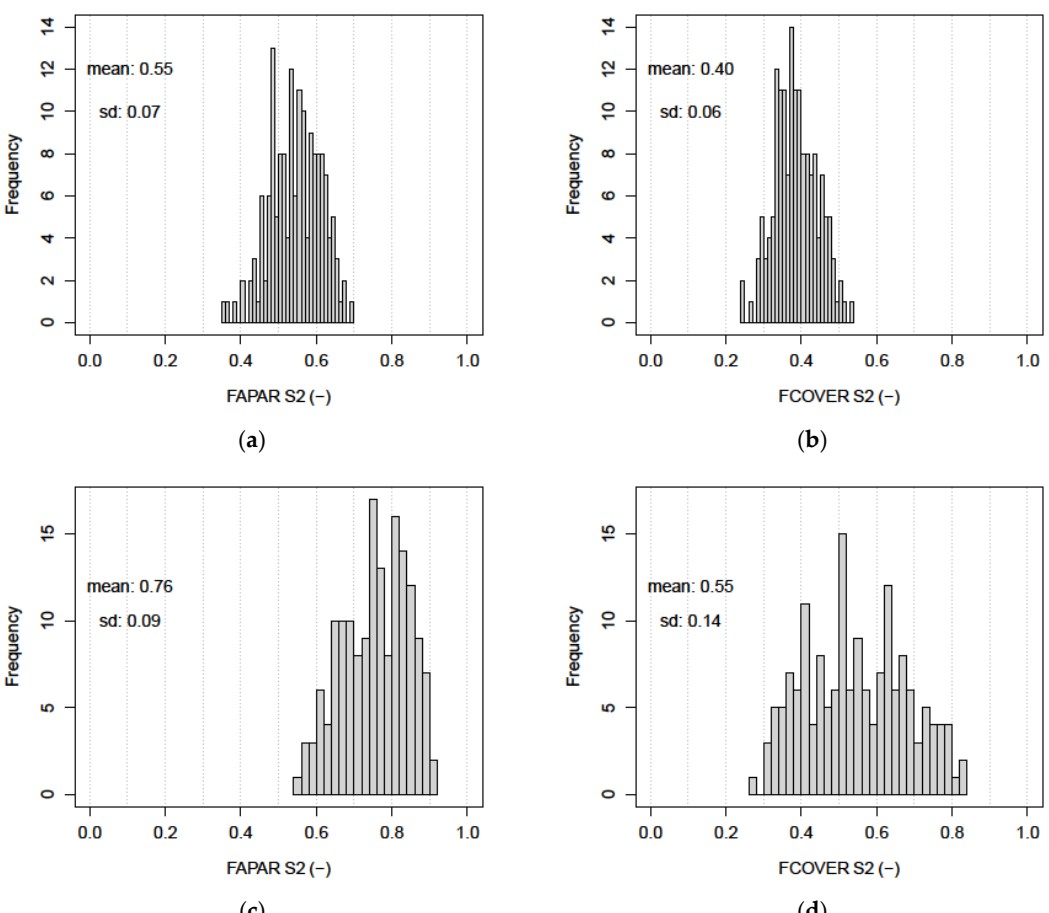

**Figure 5.** Value distributions of FAPAR-S2 and FCOVER-S2 across the study site in the spring (**a,b**) and summer (**c,d**).

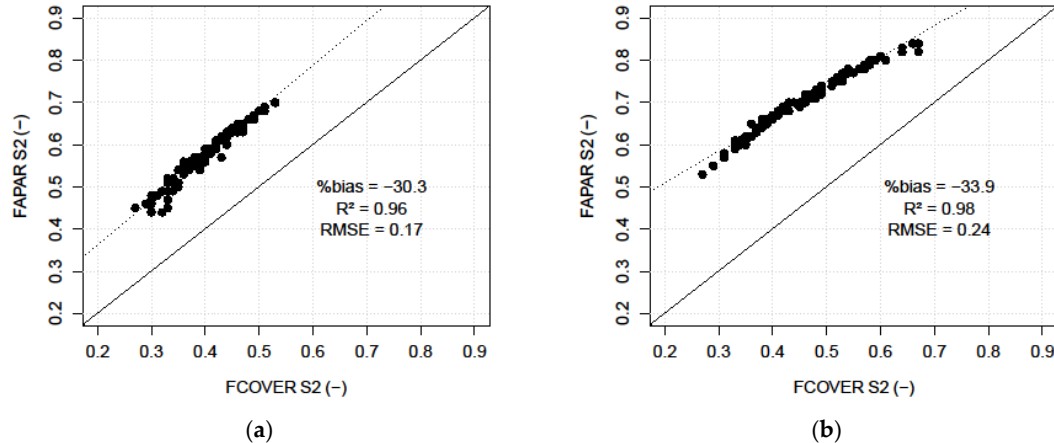

**Figure 6.** FCOVER-S2 plotted (black dots) against FAPAR-S2 across the study site in the spring (**a**) and summer (**b**).

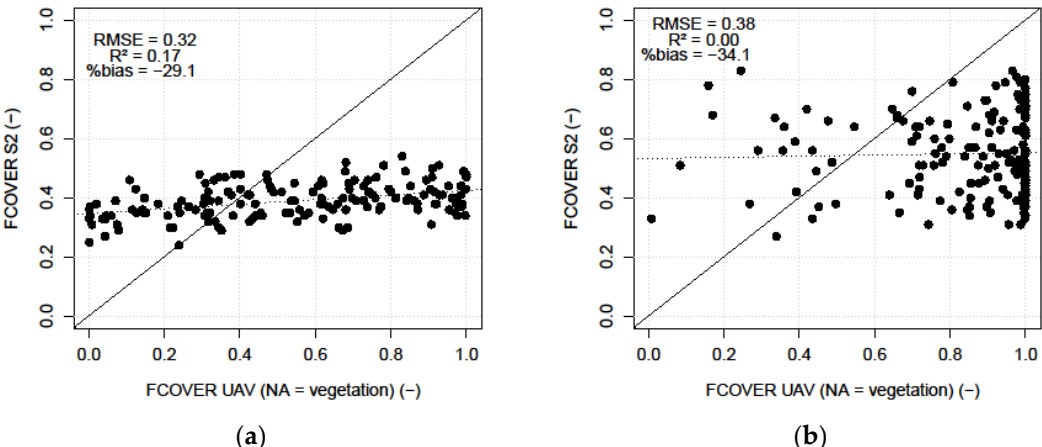

**Figure 7.** FCOVER-UAV (assumption NA = vegetation) plotted (black dots) against FCOVER-S2 at the (**a**) spring acquisition and (**b**) summer acquisition dates.

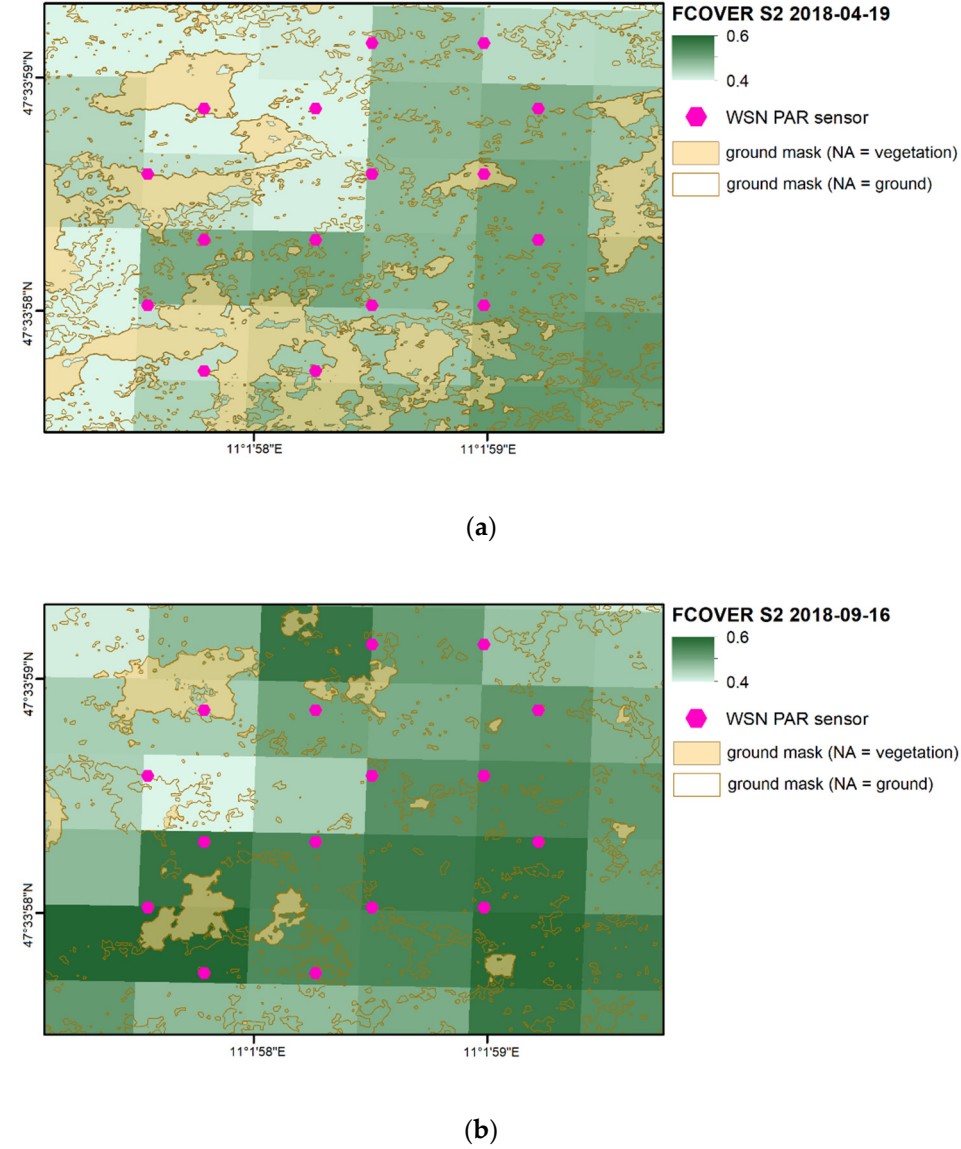

**Figure 8.** FCOVER-S2 and UAV-derived ground masks at (**a**) the spring and (**b**) summer acquisition dates.

### 3.2. Evaluation with In-Situ FAPAR

#### 3.2.1. Absolute Values and Seasonal Dynamics

The spatially distributed products, i.e., FCOVER-S2, FAPAR-S2, FCOVER-UAV$_{NA=ground}$ and FCOVER-UAV$_{NA=veg}$, were compared to FAPAR ground measurements acquired by the WSN at individual sensor locations. In this regard, Figure 9 shows a comparison of all FAPAR and FCOVER products. Most FAPAR-WSN values are consistently above 0.9 and thus higher than FAPAR and FCOVER remote sensing products for both the spring and summer acquisition dates. Contrarily, the S2 products present the lowest values, ranging from 0.5 for FCOVER in the spring to 0.8 for FAPAR in the summer. FCOVER-UAV ranges from 0.7 to 1.0, depending on the underlying ground mask and acquisition date.

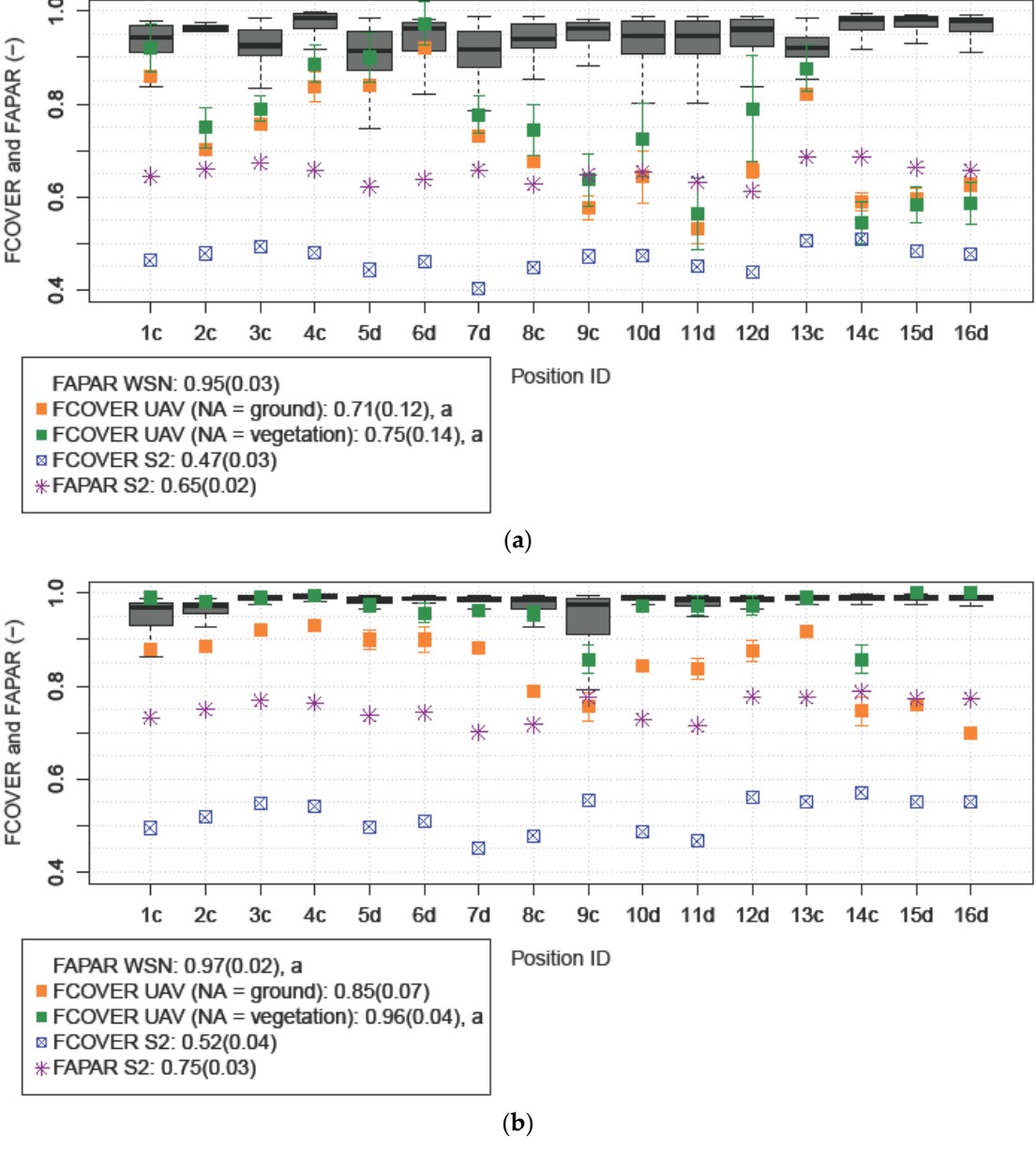

**Figure 9.** Boxplots of FAPAR-WSN and remote sensing products at the (**a**) spring and (**b**) summer acquisition dates. The legend shows the mean (sd) of all the obtained values; the results of the KS test are indicated with small letters (*p* > 0.05). Position IDs (x-axis) of the PAR sensor locations indicate species compositions in the 10-m surroundings as follows: c = ratio of basal area of conifers ≥85% and d = ratio of basal area of conifers <85%.

At the domain level, the value distributions of FAPAR-S2 and FCOVER-S2 are significantly different from FAPAR-WSN and FCOVER-UAV products for both acquisition dates (Figure 9). The FCOVER-UAV products present similar distributions in the spring but significantly different value distributions for the summer acquisition date ($p \leq 0.05$). In the summer, FCOVER-UAV$_{NA=veg}$ (mean = 0.96) is close to FAPAR-WSN (mean = 0.97) (Figure 9b), and the null hypothesis of an equal distribution cannot be rejected. In other words, FCOVER-UAV$_{NA=veg}$ and FAPAR-WSN have similar central tendencies and spatial variabilities in the summer.

Another aspect of the comparison is an evaluation of the seasonal dynamics of different products. In this regard, Figure 10 shows the difference between the summer and spring values of different products. As one would expect, FAPAR-WSN shows slightly lower seasonal dynamics at locations with higher percentages of conifers. The seasonal dynamics differ among the products: the seasonal range is the lowest for FAPAR-WSN (0.02) and highest for FCOVER-UAV products ($\geq$0.14), whereas the S2 products range in between (0.05–0.10).

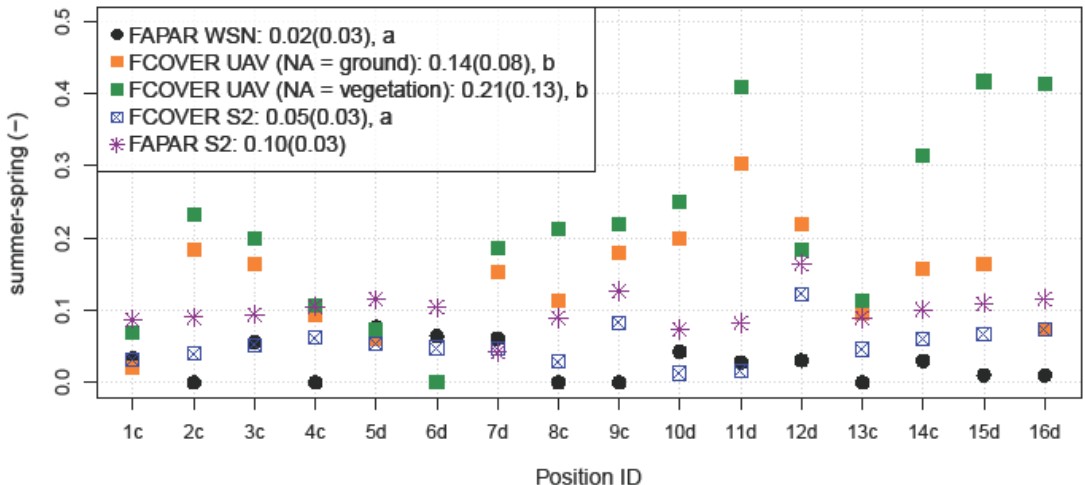

**Figure 10.** Difference between the summer and spring values of FAPAR and FCOVER products at individual sensor locations of the WSN. The legend gives the mean (sd), and the results of the KS test are indicated with small letters ($p > 0.05$). Position IDs (*x*-axis) of PAR sensor locations indicate species compositions in the 10-m surroundings as follows: c = ratio of basal area of conifers $\geq$85% and d = ratio of basal area of conifers <85%.

From the scatterplots in Figure 11, it becomes clear that there are only weak-to-moderate linear relationships between FAPAR-WSN and FCOVER-UAV at the individual sensor locations. The agreements with FAPAR-WSN are higher for FCOVER-UAV$_{NA=veg}$ than FCOVER-UAV$_{NA=ground}$, with a maximum $R^2$ of 0.51 and minimum RMSE of 0.02 at the summer acquisition date (Figure 11c). Further, the FAPAR-WSN and FCOVER-UAV products were rather uncorrelated at the spring acquisition date. Thus, the level of agreement is driven by the amount of foliage and is higher when the canopy is almost closed.

We evaluated the species compositions of the sensor locations regarding a potential difference in the FAPAR–FCOVER relationships. in the spring, locations with less conifers showed higher correlations between FAPAR-WSN and FCOVER-UAV when compared to all the locations (Figure 11a,b). Contrarily, this phenomenon did not occur at the summer acquisition date (Figure 11c,d). Coniferous and deciduous trees have distinctly different crown architecture, and the effect of species composition in combination with more open canopy seems to affect the level of agreement between FAPAR-WSN and FCOVER-UAV.

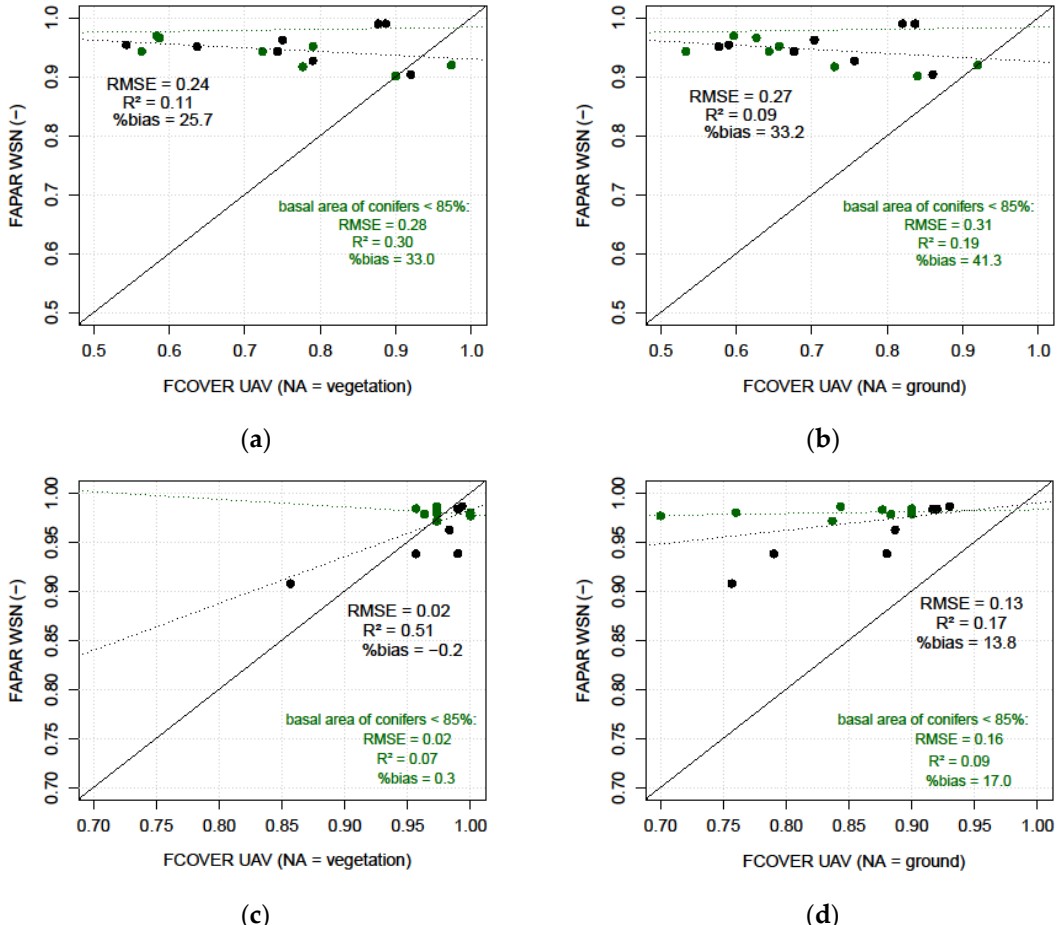

**Figure 11.** In situ FAPAR plotted against (**a**) FCOVER-UAV$_{NA=veg}$ and (**b**) FCOVER-UAV$_{NA=ground}$ at the spring acquisition date, as well as (**c**) FCOVER-UAV$_{NA=veg}$ and (**d**) FCOVER-UAV$_{NA=ground}$ at the summer acquisition date. The dot colors refer to species composition at different sensor locations (black: conifer-dominated, green: higher percentage of deciduous species).

### 3.2.2. Spatial Variability

Despite the partly high differences of the absolute values, the spatial variability of the products is relatively similar, especially at the summer acquisition date (Figure 9b). Figure 12 summarizes previous findings on spatial variability among the products by showing the maximum coefficient of variation (CV) as a function of the number of sample locations that correspond to the sensor locations for FAPAR measurements. Decreases of spatial variability due to the increasing sample size were similar across all products, which was further indicated with the high coefficients of determination ($R^2 \geq 0.95$) in Figure 13.

At both acquisition dates, FAPAR-WSN and FCOVER-UAV presented the lowest and highest spatial variability, respectively (see also the values of standard deviation in Figure 9). Similarly, both the FAPAR-WSN domain and FAPAR-S2 domain agree in a lower spatial variability compared to FCOVER-UAV and FCOVER-S2. Thus, despite their systematic underestimation, the S2 products are consistent with the differences in spatial variability observed between in situ FAPAR and FCOVER-UAV. Decreases of spatial variability of FAPAR-WSN and FAPAR-S2 are the most similar (Figures 12 and 13b), which suggests that FAPAR-S2 reflects the (relative) differences in FAPAR due to species composition and canopy cover.

The spatial variability is higher at the spring acquisition date across all products, which is attributed to the leafless deciduous species. At this time, the spatial variability of FCOVER-UAV accounts for about half in the summer compared to the spring (Figure 12). In the summer, the coefficients of variation of FCOVER-UAV and FAPAR-WSN are closely

related ($R^2$ = 0.95, RMSE = 3.52, see Figure 13a). Referring to the minimum acceptable uncertainty of 5% set by the GCOS for FAPAR products, the corresponding upper limit of the sampling size can be identified.

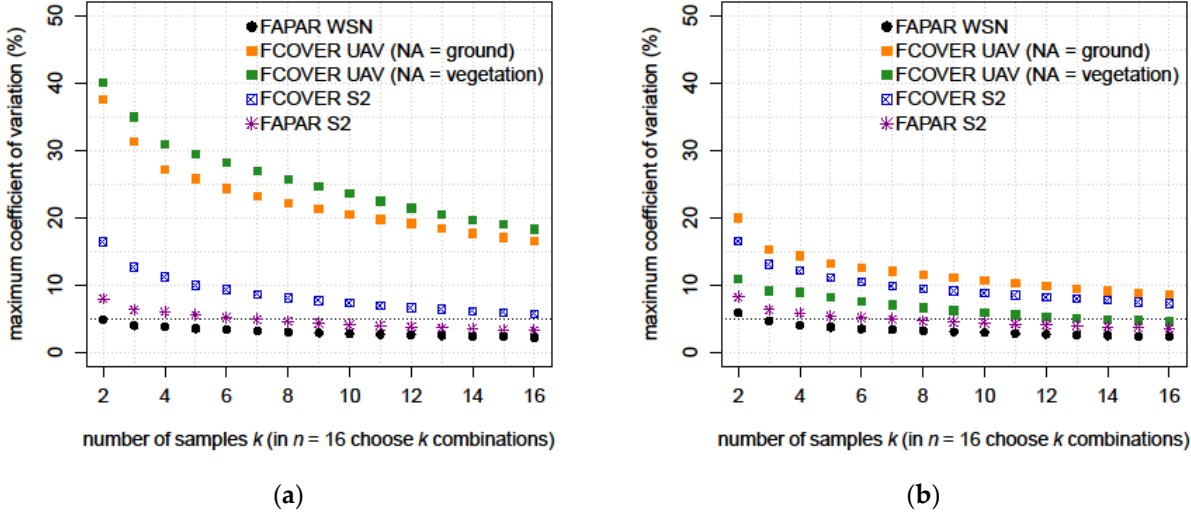

(**a**)  (**b**)

**Figure 12.** Spatial variability of FAPAR and FCOVER products shown as the maximum coefficient of variation in percent as a function of the number of samples at the (**a**) spring and (**b**) summer acquisition dates. The dashed line at 5% refers to the minimum uncertainty accepted by the GCOS for FAPAR products.

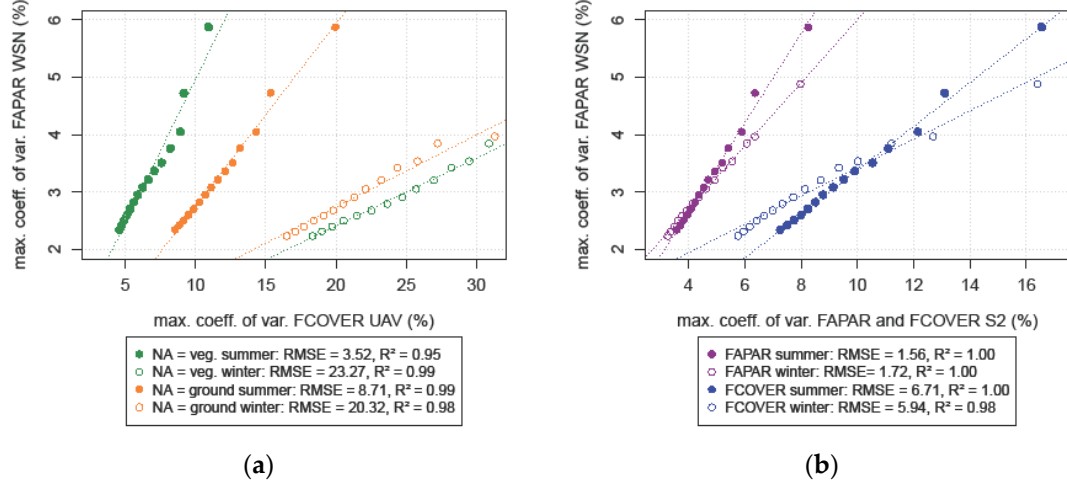

(**a**)  (**b**)

**Figure 13.** Scatterplots of the maximum coefficient of variation between FAPAR-WSN and (**a**) FCOVER-UAV, as well as (**b**) S2 products, for the spring and summer acquisition dates.

## 4. Discussion

### 4.1. Consistency of Remote Sensing Products

#### 4.1.1. UAV-Derived FCOVER

From the gap analysis, it is clear that the UAV-derived ground masks differed considerably (Table 2). The difference between FCOVER-UAV$_{NA=ground}$ and FCOVER-UAV$_{NA=veg}$ was particularly pronounced at the summer acquisition date. At this time, the canopy was almost closed, and only small gaps remained between the trees. The smaller ground area and higher occurrence of dark shadows are unfavorable for the point matching of SfM processing, so a higher amount of data gaps occurred in the dense point cloud compared to the spring acquisition date. Contrarily, similar ground masks were obtained at the spring acquisition date when sunlight could reach deeper into the forest. The problem of

dark areas leading to data gaps occurring in relatively closed forest stands is not new and represents a well-known limitation of the SfM technique [49,50]. Thus, methods exploiting the spectral information are limited at this site as well. As an alternative to SfM, it has been found that airborne laser scanning (ALS) performs better in denser canopies [68]. Clearly, SfM is better suited in less dense canopies [49]. Even though SfM approaches are efficient and the required software is well-established [49], a systematic uncertainty assessment of quality affecting factors is widely missing in single studies on UAV-derived FCOVER [57]. In this regard, we believe that the transparent handling of data gaps and the assessment of effects on retrieved FCOVER represents an important step towards the development of such processing protocols.

In sum, our results showed a consistent seasonal range of FCOVER-UAV$_{NA=ground}$ and FCOVER-UAV$_{NA=veg}$, which confirms the overall suitability of the method. Nevertheless, one should face the question of which FCOVER product could be preferable. In this regard, it is important to evaluate the ground masks required for FCOVER retrieval in accordance with the definition of FCOVER. Commonly, the definition of FCOVER does not refer to within-canopy gaps [44]. In fact, assuming NA as ground resulted in a high number of "very small gaps" (<1 m$^2$) (Table 2), of which the majority could be identified as within-canopy gaps by visual inspection (Figure 3). Hence, we consider FCOVER-UAV$_{NA=veg}$ as our reference UAV-derived FCOVER product and will therefore focus on this product in further discussion.

### 4.1.2. Sentinel-2 Products

Our results indicated a general underestimation of S2 FAPAR and FCOVER products (Figures 7 and 9). At this study site, a previous study had a validated multi-year time series of FAPAR-S2 based on direct ground observations [28]. The validation of FAPAR retrieved with the 20-m algorithm revealed a systematic underestimation of FAPAR by 25% on average [28]. In this study, we deliberately applied the newly implemented version of the algorithm that uses only 10-m bands so that we could better account for the high resolution of UAV images and check on potential improvements of the algorithm. Although a comprehensive validation study of both versions was outside the scope of this study, the comparison with the ground data confirmed again the previously found underestimation of the S2 products (Figure 9). Here, our findings are in contrast with existing studies that found good overall agreement in crops [39] and beech forest [61]. Recently, a validation of several biophysical products of S2 at several sites and biomes showed good agreement of the FAPAR product with ground data [61], but the biome "forest" was only represented by beech forest. Thus, it should be noted that there is still a lack of validation studies in different forest ecosystems and that a higher number of validation studies exist on biophysical products derived from other satellite data than Sentinel-2 [69]. It becomes clear that further ground data is required for a more representative validation of the S2 products in forest ecosystems.

Potential reasons for the high discrepancies of absolute values at this site have been discussed before, with the generic nature of the algorithm and the higher sensitivity of direct FAPAR estimates to account for the spatial variability of the forest light environment [28]. Further, the site is characterized by relatively high ecosystem complexity due to the mixed-coniferous forest with different crown geometries. In addition to that, the findings of this study led to additional explanations. Interestingly, the overlay of UAV-derived ground masks with FCOVER-S2 (Figure 8), as well as the quantitative assessment between FCOVER-UAV and FCOVER-S2, revealed better agreement in the spring (Figure 7). The seasonal difference in product agreement could arise from an effect of spectral mixing due to the coarser spatial resolution of S2 compared to the centimetric resolution of the UAV imagery. A higher influence of spectral mixing between ground and canopy features was presumed at the summer acquisition date when the canopy was almost closed, leading to the observed loss of correlation between FCOVER-UAV and FCOVER-S2 at this time. At the spring acquisition date with leafless deciduous canopy, a higher percentage of ground and bigger

gaps in the canopy were present, so the effect of spectral mixing was expected to be less prominent. Another reason for the observed loss of correlation between FCOVER-UAV and FCOVER-S2 in the summer could be attributed to the spectral signature of the understory vegetation. At the spring acquisition date, the understory vegetation was dormant, whereas, in the summer, the forest floor was covered almost completely by herbal understory (Figure A1). It is important to stress that the S2 algorithm relates to the green elements of vegetation only [17], so a green understory may represent a potential caveat, meaning that the algorithm is not able to interpret and distinguish between forest floor and canopy at this site. The phenology of the understory may also explain why the seasonal range of the S2 products was widely different to the seasonal range observed in FCOVER-UAV (Figure 9), which was based on the elevation instead of spectral information. In sum, it is questionable whether it is possible to accurately retrieve FCOVER with a generic algorithm and optical remote sensing at this site. At least for FCOVER, optical remote sensing with a higher spatial resolution than S2, preferably in the range of decimeters, or even other space-born products based on Lidar technology such as the GEDI instrument [70], could be more appropriate, but spatial coverage, multitemporal data availability and spatial resolution limit its suitability for monitoring single forest stands.

Compared to the accuracy of the absolute values, the S2 products demonstrated relatively high consistency (Figure 6). First, FAPAR-S2 was consistently higher than FCOVER-S2, which was also the case for FAPAR-WSN compared to FCOVER-UAV (Figure 9). Second, the spatial variability of FAPAR-S2 was consistently lower than the spatial variability of FCOVER-S2, which again coincided with observations of FAPAR-WSN and FCOVER-UAV (Figure 12). Regarding the FAPAR product, previous investigations have already pointed out the sensitivity of the FAPAR-S2 to map differences of the species compositions [28], and this is now also confirmed for the FCOVER product at this site. Thus, our findings suggest a general capability of S2 and the algorithm to detect seasonal changes and spatial variability of FAPAR and FCOVER in forests.

### 4.2. FCOVER as a Proxy for FAPAR

#### 4.2.1. Influence of Spatial Resolution and Acquisition Conditions

A careful consideration of product specifications related to the spatial scale and acquisition conditions is required for evaluating the suitability of FCOVER as a proxy for in situ FAPAR. The very high spatial resolution of UAV imagery accounts for a detailed characterization of canopy gaps and thus allows for a spatial intersection with in situ FAPAR at the centimetric scale. It should be considered that product comparability is dependent on the geometric accuracy of the datasets. For example, the horizontal error of GNSS positioning for the ground control points reached up to 0.5 m. In situ FAPAR is known to be highly variable in forests, and geometric inaccuracies may explain why only a moderate relationship was achieved between FAPAR-WSN and FCOVER-UAV in the point-to-point comparisons (Figure 11). Nevertheless, the low bias and high agreement of the domain values suggest an acceptable spatial congruency of the datasets.

Regarding the satellite-derived FCOVER product, the decametric resolution of S2 allowed for a comparison with ground measurements based on single sensor locations. As discussed before, the evaluation of the spatial variability demonstrated the capabilities of S2 to reflect different species composition at this site. However, in this dense forest stand with numerous canopy gaps less than one square meter (Table 2), the 10-m resolution of S2 inevitably leads to the mixing of canopy and ground area. In this regard, the mixing of spectra signatures from the top-of-canopy and ground levels may lead to a homogenization of FCOVER values and thereby explain the more centered value distributions of the S2 products compared to the larger value range observed for UAV-derived FCOVER (Figures 4 and 5). Here, our results are in accordance with the observations of Riihimäki et al. [51], who discussed the effect of spatial resolution on FCOVER in tundra vegetation in the context of the classical modifiable area unit problem (MAUP) in geoinformation data [71].

Apart from the differences in spatial scale, the acquisition conditions and related effects on product accuracy need to be considered. Acquisition conditions for remote sensing products may vary regarding the solar position and illumination conditions. The solar azimuth and zenith angles during the acquisition times were quasi-identical to the two UAV flights, so the comparability of the UAV-derived FCOVER was ensured. Similar acquisition conditions could also be assumed for the S2 images, since their acquisition time was at 11:25 local time, i.e., approx. one hour earlier than the UAV flights. Regarding the illumination conditions, a general (temporal) conflict between the optimal acquisition conditions among the remote sensing products should be noted. While the S2 products are available for cloudless conditions only, it is often recommended to carry out UAV campaigns during overcast conditions to limit the occurrence of dark, shadowed areas and thus increase the quality of dense point clouds [49]. In principle, the permanent FAPAR observations allowed scheduling the UAV flights whenever the meteorological conditions were suitable. In practice, frequently higher wind speeds and prolonged snow cover in the spring, as well as frequent thunderstorms in the summer, restricted UAV flights at the subalpine site. Our initial intent to perform the UAV flights under overcast conditions was not feasible in the spring due to a short time window between complete snow melt (which is important for the accuracy of in situ FAPAR) and the leaf-sprouting of beech trees. At that time, only days with clear sky conditions allowed for UAV flights, so the shadows in the imagery could not be avoided.

For future research, it could be worthwhile to use UAV imagery acquired under overcast conditions. However, it should then be considered that S2 products relate to clear sky conditions only, thereby complicating the validation against in situ FAPAR, which is known to be significantly affected by different illumination conditions [35,36]. In the spring, we acquired the UAV flight on the same day of a S2 overpass, and at the summer acquisition date, the S2 data was acquired only three days later than the UAV imagery, when there was still no leaf development according to an automated camera (Figure A1). Thus, even though the quality of the UAV imagery may be compromised due to the occurrence of shadows, we ensured that all products referred to clear sky conditions only so that both the absolute values and spatial variability of FAPAR and FCOVER could be evaluated.

### 4.2.2. Conceptual Differences between FAPAR and FCOVER

FAPAR and FCOVER are different quantities. Whereas FCOVER values are driven by the vertical projection of the canopy only, FAPAR is driven by the scattering and absorption of PAR and the vertical alignment of vegetation elements. An approximation of in situ FAPAR by FCOVER relates to the sub-concept of FIPAR, which assumes that solar radiation is either absorbed by the canopy or passes through gaps onto the ground [36]. Our results indicate that the suitability of UAV-derived FCOVER as a proxy for in situ FAPAR was dependent on the season. In the summer, a moderate relationship ($R^2 = 0.50$) and high overall agreement with low RMSE between FAPAR and FCOVER-UAV was observed. According to Gobron et al. [45], FIPAR can be approximated by FCOVER with errors of typically around 0.1. While we observed deviations clearly below this value in the summer (<0.02), higher deviations were found for the spring acquisition date (~0.26). In more open canopy, as present for the spring acquisition date, light can penetrate from various angles through the canopy, leading to multiple scattering and absorption, and these processes only affect FAPAR. In dense canopy, light cannot penetrate through the lateral sides of the canopy and will therefore mainly be absorbed or reach the ground. Thus, FCOVER-UAV was a good proxy for absolute values of FAPAR only in the summer. In addition, FAPAR showed a lower spatial variability than FCOVER. In fact, we found the variability of UAV-derived FCOVER to be twice the variability of the measured FAPAR.

The presence of different species offered the possibility to further analyze the FAPAR–FCOVER relationship according to species composition. It has been observed that, in the spring, sensor locations surrounded by a lower ratio of spruces (in terms of basal area) showed better agreement. It should be noted that this observation relies on only a few

samples ($n$ = 16) and that the basal area may not correspond linearly to the actual crown area of both individual trees and the different tree species. Nevertheless, it is possible that the distinct differences in crown architecture, clumping and, thus, scattering properties of needleleaf and broadleaf species alter the FAPAR–FCOVER relationship. Conifers are able to intercept incoming PAR from a larger range of solar elevation angles [72], which may lead to proportionally higher values of FAPAR despite relatively low FCOVER at a given location. Based on our results, we would expect a better relationship between FAPAR and FCOVER when the solar zenith angle is the lowest and at forest stands with broadleaf canopies. In this regard, future studies could investigate the FAPAR–FCOVER relationship in purely broadleaf or needleleaf forest stands. Nevertheless, it should be considered that mixed forests like the one at our site could play a crucial role in (European) forestry due to their increased drought resilience [73], so studying FAPAR in mixed stands is likely to gain further scientific interest.

In addition to tree species, structural parameters such as the tree height, basal area and crown geometry may also influence the FAPAR–FCOVER relationship. However, due to the complexity of scattering processes in forests [36], it is known that the relationship between forest structure and light environment is not linear. It has been shown that at least common forest structure variables such as stem density, basal area, tree height or diameter at breast height (DBH) are not always appropriate predictors of light availability in a forest [34]. Again, the importance of certain structural parameters for the FAPAR–FCOVER relationship may be species-dependent. For example, Norway spruce with narrow shapes has been found to exhibit a higher needle density per unit of crown volume and higher leaf index compared to broad-crowned trees [74]. For European beech, the horizontal extension of the crown has been identified as the main driver of tree growth [75]. Studying the influences of the structural parameters was beyond the scope of this study and not feasible due to the stand's similar age structure. Future studies, preferably theoretical approaches based on radiative transfer modeling, could further address the influence of structural parameters and tree architectures.

### 4.2.3. Optimizing FAPAR Sampling with UAV-Derived FCOVER

Our experiment revealed that the spatial variability of FAPAR did not exceed the spatial variability of FCOVER. In the context of the validation of FAPAR and FCOVER products, we suggest that the spatial variability of UAV-derived FCOVER could be used as an orientation for setting an appropriate sampling size for ground measurements. Since the FCOVER products exhibited higher spatial variability than FAPAR products, the spatial variability of FCOVER represents a benchmark for the upper limit of the spatial variability of FAPAR. We therefore suggest using UAV-derived FCOVER to estimate an appropriate sampling size for direct FAPAR measurements. Specifically, the associated sampling size with a minimum permissible coefficient of variation can be read from the course of FCOVER-UAV in Figure 12. If one refers to the 5% threshold, the upper limit corresponds to a maximum of 12 repetitions of direct FAPAR measurements in dense canopy.

Apart from direct FAPAR measurements, as carried out in this study, indirect methods can also be optimized with this approach. Since previous studies have shown high correlations of UAV-derived FCOVER with a gap fraction obtained from DHP [57], and validation studies of FAPAR mostly rely on DHP [69], FAPAR sampling based on DHP could also benefit from a priori information on UAV-derived FCOVER.

## 5. Conclusions and Recommendations

FAPAR measurement campaigns in forests are regarded as cost- and labor-efficient, with the consequence that datasets available for the validation of satellite-derived FAPAR products are scarce. At the same time, the rapid technical progress and availability of UAVs in combination with the development of evaluation routines to retrieve forest structural parameters inspired us to use UAV-derived FCOVER for the FAPAR quality assessment. We assessed the potential of remote sensing-derived FCOVER as a proxy for in situ FAPAR

in a dense mixed-coniferous forest in the context of validating satellite-derived FAPAR products. We therefore evaluated in situ FAPAR based on direct PAR measurements, UAV-derived FCOVER and the S2 FAPAR and FCOVER products in a dense mixed-coniferous forest, ensuring almost identical illumination conditions (i.e., solar position and clear sky conditions) at their acquisition times.

Our investigation on absolute values and the spatial variability of FAPAR and FCOVER yielded two main results. First, similar value ranges between UAV-derived FCOVER and in situ FAPAR were observed in the summer when the canopy was almost closed. We further observed a systematic underestimation in the S2 FCOVER product as observed in previous investigations on FAPAR at this site, which we mainly retraced to the concept of the generic retrieval algorithm and the coarser resolution of the product. We argued that the conceptual differences of in situ FAPAR and very high-resolution UAV-derived FCOVER products are negligible if the majority of the incoming PAR is either absorbed or passes through gaps in the canopy. As a second main result, we found that, in all products, the spatial variability of FCOVER outranged that of FAPAR, which can be retraced to the conceptual differences between both quantities.

Our experimental findings pointed to the general conclusion that FAPAR sampling and validation activities can strongly benefit from UAV-derived FCOVER. First, the expected range of values of in situ FAPAR may be estimated across a dense stand (i.e., referring to FCOVER above 0.8) with a priori information on the FCOVER. As the domain-level FAPAR in our dense forest stand could be approximated via UAV-derived FCOVER, this could also imply that permanent FAPAR monitoring systems should be targeted at more open stands or areas undergoing succession after forest disturbance. Further, to facilitate FAPAR sampling and refine FAPAR sampling protocols, we recommend using UAV-derived FCOVER to estimate the upper limits of the sampling size for in situ FAPAR measurements. While it is commonly accepted that multiple FAPAR measurements are required in forests, assessing the maximum sampling size as a practical aspect in sampling efforts has remained obscure. In this regard, we believe that the development of FAPAR sampling protocols could benefit from our proposed approach based on UAV-derived FCOVER, following the GCOS accuracy requirements.

Given the importance of forests for the global terrestrial carbon balance, we expect the remote sensing of FAPAR to gain importance in forestry and land use decision-making also at smaller spatial scales. Until this can be reached at the operational level, satellite products must be continuously improved, which can be achieved by independent reference data but also via intercomparisons of biophysical parameters retrieved from the same sensor. In this regard, we recommend including our findings on the different spatial variabilities between FAPAR and FCOVER products as consistency criteria. In addition, the seasonally increased the spatial variability of FAPAR, and FCOVER could be formulated as another consistency criterion for assessments in mixed and deciduous forests. In sum, we encourage future quality assessments to be conducted with respect to the spatial variability of FAPAR in different forest ecosystems, thereby enabling a more comprehensive understanding of the actual capabilities of FAPAR remote sensing.

**Author Contributions:** B.P. performed the FAPAR measurements, conceptualized the study and developed the methodology, processed, and analyzed all the data and wrote the original draft of the manuscript. P.M. planned and carried out the UAV flights, processed the UAV data and contributed to the methodology. P.K. contributed to the conceptualization and methodology of the study. A.S.-A. conceptualized the FAPAR measurements and acquired the funding. All authors have read and agreed to the published version of the manuscript.

**Funding:** This work was supported by funding from Helmholtz Association and the Federal Ministry of Education and Research (BMBF) in the framework of TERENO (Terrestrial Environmental Observatories) (Grant No. 01LL0801B). The WSN technology was provided by the University of Alberta, Edmonton, CA, receiving funding from the National Science and Engineering Research Council of Canada (NSERC)—Discovery Grant Program and Canada Foundation for Innovation.

**Institutional Review Board Statement:** Not applicable.

**Informed Consent Statement:** Not applicable.

**Data Availability Statement:** The replication data for this study will be made available via the Dataverse Project (https://dataverse.harvard.edu/dataverse/Tropi-Dry, accessed on 11 January 2022).

**Acknowledgments:** The authors thank Ralf Ludwig and Ralf Kiese for the continuous support and advice.

**Conflicts of Interest:** The authors declare no conflict of interest.

## Appendix A

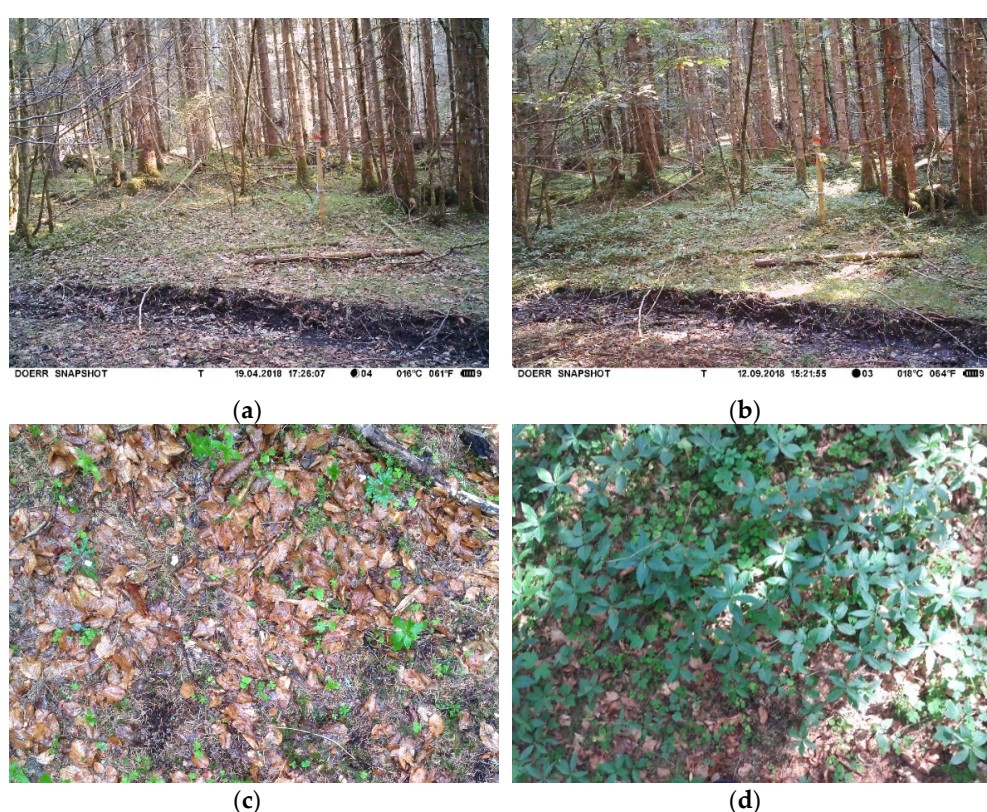

|       |       |
|-------|-------|
| (**a**) | (**b**) |
| (**c**) | (**d**) |

**Figure A1.** Photos of the forest stand with (**a**) the PAR sensor and (**b**) understory vegetation at the spring, as well as (**c**) the PAR sensor and (**d**) understory vegetation at the summer acquisition date.

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
