# Peer review of "Fractional Vegetation Cover Derived from UAV and Sentinel-2 Imagery as a Proxy for In Situ FAPAR in a Dense Mixed-Coniferous Forest?"

_remotesensing, doi:10.3390/rs14020380_

Round 1
Reviewer 1 Report
The paper investigate FCOVER as a proxy for FAPAR in a dense mixed-coniferous forest. This is an important topic as findings may facilitate the development of future FAPAR using Remote Sensing. The manuscript is scientifically sound and reads well. I present below some comments to be considered before accepting the paper.
Line 193-195, what is the Yao’s mehtod. The author said that they used elevation information of the point cloud for the retrieval of ground masks. What is the threshold, please clarify.
Compared with UAV, sentinel-2 has lower spatial resolution. This may cause mixed pixel problems (A pixel contains the ground and trees). Please discuss the effect of mixed pixel on FAPAR and FCOVER.
What is the effect of tree height on FAPAR when the FCOVER is the same?
the conclusion can be further improved.
Author Response
Dear Reviewer, please see our response in the attachment.

Reviewer 2 Report
FCOVER as a proxy for FAPAR in a dense mixed-coniferous forest?
This manuscript investigates fractional vegetation cover (FCOVER) as a proxy for the fraction of absorbed photosynthetic active radiation (FAPAR) in a dense mixed-coniferous forest, assessing absolute values and spatio-temporal variability of in-situ FAPAR and FCOVER derived from UAV and Sentinel-2 imagery. The findings from this research can be included as consistency criteria for quality assessment of FAPAR and FCOVER derived from other remote sensing products.
The topic of this study is acutal and very interesting for the readers, however, still the manuscript needs some minor corrections. My minor concerns are, as follows:
- In my opinion, using two abbreviations in a Title (although they are well-known) does not contribute to the accessibility of the scientific paper. Perhaps the Authors should consider of adding used sensors in this research?
- Section Introduction is very well written. However, last paragraph does not fit in this Section. LN 104-110 indicate the main objectives of this research, and then it seems like LN 111-117 are inserted afterwards, e.g. LN 111 repeats with LN 120, etc.
- Please adjust terms Fig. and Figure
- Section Conclusions should be rewritten in a more concise way. Reason for writing this manuscript (e.g., first sentence of this Section) is not necessary in this Section. It should present the outcome of the work by interpreting the findings at a higher level of abstraction than the Discussion and by relating these findings to the motivation stated in the Introduction.
Overall, I think that the research design of this manuscript is very good and actual, but still some additional changes need to be made.
Author Response

(The authors gave the same response as above.)

Reviewer 3 Report
The abstract was written in a hurry and should be improved. Please see my comments

Author Response

(The authors gave the same response as above.)

Reviewer 4 Report
Abstract should be improved in such way as to underline the most important results. The last paragraphs of current form sound more like conclusions.
Author Response
Dear Reviewer,
We would like to thank you for your time to read our manuscript and your suggestions for further improvements.
Following your advice, we have thoroughly revised and expanded the abstract to provide more details on our results before stating the conclusions, see the revised version of the manuscript.
Again, thank you for your advice on how to improve the manuscript.
Round 2
Reviewer 1 Report
Thank you for taking the time to respond to all my questions and comments.
Overall, I am satisfied with the responses of the authors and have no qualms recommending the manuscript for publication in its current, significantly improved, form.
Good luck!